# Influence of Airline Cabin Crew Members’ Rapport-Building Behaviors and Empathy toward Colleagues on Team Performance, Organizational Atmosphere, and Irregularity

**DOI:** 10.3390/ijerph18126417

**Published:** 2021-06-13

**Authors:** Jungyi Park, Sunghyup Sean Hyun

**Affiliations:** School of Tourism, Hanyang University, 222, Wangsimni-ro, Seongdong-gu, Seoul 04763, Korea; jungyi6262@hanyang.ac.kr

**Keywords:** rapport-building behavior, uncommonly attentive behavior, common-grounding behavior, courteous behavior, connecting behavior, information-sharing behavior, empathy among cabin crew, team performance, organizational atmosphere, irregularity

## Abstract

Expanding on the literature on rapport-building behavior within the airline industry, this study analyzed the influence of rapport-building behaviors (uncommonly attentive behavior, common-grounding behavior, courteous behavior, connecting behavior, and information-sharing behavior) on cabin crew members’ empathy toward their colleagues. We also analyzed the effect of empathy on variables such as team performance, organizational atmosphere, and instances of irregularity. We analyzed 230 samples obtained from an online questionnaire and convenience sampling of full-service domestic and international carriers in South Korea. A structural equation modeling (SEM) revealed that uncommonly attentive behavior, courteous behavior, connecting behavior, and information-sharing behavior showed a positive effect on empathy among colleagues, which in turn positively influenced team performance, organizational atmosphere, and possible irregularities. Moreover, we found that the presence of participants’ closest colleagues within the same team did not moderate the relationship between rapport-building and empathic behavior between airline crew members. Our study has important implications for crew members’ dignity and protection from emotional labor while working in high-pressure environments. Our findings can be used to revise the airline industry’s crew management guidelines and improve the crew’s psychological health and quality of life.

## 1. Introduction

To date, in South Korea, there has been a constant interest regarding ways to improve airline cabin crew’s performance and the services offered to the customers. However, most recent studies have focused on ensuring the human rights of individual cabin crew members and on other psychological and emotional aspects of the job [1,2,3].

The purpose of this study is to confirm the effectiveness of rapport-building behavior among cabin crew members in enhancing their cooperation with each other and increasing the synergy effect achieved.

A literature review revealed that rapport-building behavior strengthens the psychological trust between the service provider and the customer, and additionally, it improves the quality of interaction between them [4]. Furthermore, the communication between the leader and members, i.e., “leader–member” communication, is viewed as an extremely important factor. Therefore, a method that leaders can utilize to improve the quality of the relationship was presented, and a model of leadership communication was suggested based on the rapport management theory [5]. Additionally, there are studies that report that the more empathetic the leader is, the higher the member’s job satisfaction is likely to be [6]. Therefore, this study expanded on previous research and focused on the relationship among cabin crew members.

However, few studies have focused on rapport-building behaviors and empathy among cabin crew members. In this study, we considered aspects such as airline workers’ working conditions and dignity of labor, and expanded on the concepts of rapport and empathy in the context of basic human relationships with respect to airline cabin crew members.

Cabin crew members in Korea’s domestic airlines work as a team. A few previous studies show that psychological bonds are important for rapport building, which is crucial for cabin crew members who always work in close cooperation. Therefore, this study also focused on rapport-building behaviors.

Based on a comprehensive literature review, we classified rapport-building behavior between cabin crew members into five categories: (1) uncommonly attentive behavior between the crew members, (2) common-grounding behavior, (3) courteous behavior, (4) connecting behavior, and (5) information-sharing behavior [7,8,9,10]. Then, we investigated the behaviors that had the strongest effect on empathy among the crew members.

Since a cabin crew works within a group/team, a psychological bond between colleagues is essential to help the members intuitively identify instances when their colleagues experience difficulties during their work and the type of assistance they may require. Academics have defined this type of psychological bond as empathy [11,12,13]. Empathy refers to a strong, deep psychological connection between two individuals, which can be formed through smooth communication (rapport building) [14,15]. In this study, it was found that rapport-building behavior has a positive effect on empathy.

Previous studies have reported that strong empathy with colleagues can have a positive effect on team performance [16,17,18] and organizational atmosphere [19,20,21], while preventing possible irregularities among employees [22,23,24]. In this study, we investigated the causality of the relationship between empathy and team performance, organizational atmosphere, and employee irregularity. We also performed a statistical analysis of the mediating role of having one’s close colleagues within and outside the same team, in the relationship between rapport-building behavior and empathy among colleagues.

In previous studies, the focus on rapport-building behavior was limited to the relationship between the employee and the customer. However, this study makes a theoretical contribution as well in that it broadens the target group for research by focusing on the rapport-building behavior among cabin crew members, which is an extension of previous research that focused only on the relationship between the employee and the customer.

No previous research exists on rapport-building behavior and empathy among cabin crew members. Therefore, this study is unique in that it is the first to set cabin crew members as the target group for research. In addition, it is academically significant as it analyzed which of the five factors of rapport-building behavior had the strongest effect on engendering empathy among colleagues. It differs from previous studies in that it concludes that if the positive interaction between crew members continues, then effects of individual, psychological, economic, and social benefits may be achieved from the perspective of cabin crew members as well as the company. This finding will be the cornerstone for human resource management planning and its improvement; in the creation of a positive work environment; and in the improvement of mental health and quality of life, work–life balance, and organizational management.

### 1.1. Literature Review

#### 1.1.1. Rapport-Building Behavior

The word rapport originates from French and refers to a relationship of trust between people. It is a comprehensive term for synchronization, wherein mutual communication transcends language, and occurs at the physiological or psychological level while also entailing a mirror effect, where individuals subconsciously mimic one another’s behavior [25,26].

After Freud, several authors have offered their own definitions of rapport, particularly in the business context. For example, rapport could imply a perception of mutual connection between staff and clients [27], or an intimate relationship formed on the basis of trust while providing a service [28]. One author defined rapport as the relationship between a service provider and a client based on experiencing positive, amicable emotions from mutual interactions [29], while others have described it as a relationship that creates an atmosphere of mutual understanding and trust [15], and a behavioral element aimed at improving the quality of the relationships between service providers and clients [4]. Drawing on the common elements from previous definitions, this study defines rapport as the formation of trust, camaraderie, and a psychological relationship between the members of an airline cabin crew interacting with each other.

Rapport is an important factor affecting economic outcomes, as developing rapport can improve customer loyalty [30,31,32,33]. At an interpersonal level, rapport-building involves increasing comfort and intimacy by expressing attention through various ways such as eye contact, control over one’s breathing, posture, movements, and other forms of body language. However, social rapport has a positive effect on the satisfaction of other parties [34].

Previous studies have classified rapport-building behavior into five types: uncommonly attentive behavior, common-grounding behavior, courteous behavior, connecting behavior, and information-sharing behavior [7,8,9,10]. In this study, we analyzed these behavioral types to measure the rapport-building behavior of cabin crew members with their colleagues.

Previous studies have also shown that rapport-building behavior can create empathy among colleagues. For instance, information-sharing behavior and common-grounding behavior by tour guides have been shown to have a positive effect on clients’ empathy [9], while the use of rapport-building techniques during interpersonal conflict has positive effects on empathy and the willingness to resolve conflicts [9]. In addition, several studies have reported that personalization and listening are among the four effective rapport-building techniques, which positively impact rapport building between female investigators and government officials [14]. Accordingly, the following subsections outline the behavioral types and present our hypotheses.

##### Uncommonly Attentive Behavior

Uncommonly attentive behaviors can be divided into three subtypes [35]. The first subtype is an atypical behavior, which involves making the other party recognize that they are receiving unexpected special treatment [7]. The second subtype is personalized rapport-building, which stems from remembering the name or personal information of a specific client [36]. The third subtype is sincere personal attention [8]. For example, cabin crew members may make it a point to remember their colleagues′ birthdays and serve birthday cakes at overseas stations. When they meet on the next flight, they may remember their colleagues’ names and present them with drinks or gifts. Therefore, we hypothesized:

**Hypothesis** **1** **(H1):**
*Uncommonly attentive behavior by cabin crew members would make them more empathic toward their colleagues.*


##### Common-Grounding Behavior

Common-grounding behaviors occur when individuals try to find mutual areas of interest. They can be divided into two subtypes [37]. The first involves looking for similarities with a coworker regardless of differences in sales or performance levels. Two parties who identify their similarities and perceive each other as being similar tend to build rapport and make their relationship more robust [38,39]. The second involves the act of looking for mutual interests beyond superficial similarities [40]. For example, there may be such colleagues who receive the initial training together, advanced customer service training together, or return-to-work training together following maternity leave. In addition, there are several in-house clubs and societies where acquaintances may be formed based on common interests, which may blossom into lifelong reliable relationships. Hence, we proposed the following hypothesis:

**Hypothesis** **2** **(H2):**
*Common-grounding behavior by cabin crew members would make them more empathic toward their colleagues.*


##### Courteous Behavior

Courteous behavior implies earnestly treating the other party with interest and respect. Choi’s study showed that exhibiting courteous behavior aided rapport building [41].

Courteous behavior can be divided into three subtypes: the act of sharing empathy with a client, which is considered to be an important factor in increasing intimacy and rapport-building [36,42]; having respectful manners, [43,44,45]; and honesty, which includes displaying a friendly attitude and greeting the client. During problematic scenarios, sincere apologies and behaviors aimed at restoring the relationship have been claimed to induce positive rapport building [46]. For example, cabin crew members are courteous toward their colleagues. They may greet each other with bright smiles in pre-flight briefing rooms, nod as they walk past each other in the plane aisle, and exchange warm greetings in the galley during flights. Hence, we proposed the following hypothesis:

**Hypothesis** **3** **(H3):**
*Courteous behavior by cabin crew members would make them more empathic toward their colleagues.*


##### Connecting Behavior

Connecting behavior relies on setting clear intentions to develop a relationship [47], which is exemplified through actions such as pleasant and humorous conversations and speaking on familiar terms [48]. Connecting behavior can be divided into three subtypes [7]. First is a pleasant conversation with a client. The second is speaking on familiar terms. Connecting behavior that creates an atmosphere of entertainment or interest has been reported to help form strong bonds [7]; therefore, the third subtype involves humor [35]. For example, as cabin crew members work in teams or groups, they may hold humorous conversations with new colleagues. This will help lighten the mood and facilitate quick friendships, thereby building rapport in a short amount of time. In turn, this will be helpful in their in-flight work requiring cooperation. Hence, we proposed:

**Hypothesis** **4** **(H4):**
*Connecting behavior by cabin crew members would make them more empathic toward their colleagues.*


##### Information-Sharing Behavior

Information-sharing behavior includes behaviors such as providing clear information, sharing knowledge, and providing overall advice about a service [7]. Information-sharing behavior makes the clients feel that they are receiving a differentiated service, and both the service provider and client form a relationship based on amicable emotional exchange [49]. As part of this behavior, providing a service establishes open communication with the client [47], which also improves transparency. For example, cabin crew members may share their colleagues’ latest flight information and predict service items on their flight routes in advance. They may also share information on pre-flight preparation, exercise management, and even overseas travel knowledge. They may share information on the company′s latest announcements, and they may even study together for regular safety training sessions. Hence, we proposed the following hypothesis:

**Hypothesis** **5** **(H5):**
*Information-sharing behavior by cabin crew members would make them more empathic toward their colleagues.*


#### 1.1.2. Empathy toward Colleagues

Empathy can be divided into (a) cognitive empathy, which is the process of understanding others’ behavior and adopting their roles, and (b) emotional empathy, which entails experiencing others’ emotions vicariously [50]. Empathy enhances one’s understanding of others and is an essential factor for forming positive interpersonal relationships [51]; therefore, for an airline cabin crew, it is a core element of relationships among colleagues.

Empathy can be evoked in various work interactions, such as when a colleague expresses emotions about a difficult problem, while looking after a colleague, or acting gently [52,53,54]. Colleagues responding to each other’s problems exemplifies empathy within the group [55]. Being an essential part of relationship-building, empathy also increases intimacy among colleagues [56]. Moreover, empathy toward colleagues promotes a positive transformation of negative emotional experiences that occur within a group [57,58]. Empathy also increases group involvement and positive emotions and attitudes while reducing personal distress [59].

Many studies have reported that empathy toward one’s colleagues significantly impacts organizational effectiveness, whereas empathy from leaders positively affects the performance and failure tolerance of their subordinates. It also promotes group cohesion and work satisfaction [60]. Additionally, several studies have reported a positive correlation between the leader’s empathic ability and the member’s job satisfaction [6].

However, few studies have focused on airline cabin crew members’ perception and experience of empathy from their colleagues. Therefore, to fill this gap, this study provides a thorough analysis of cabin crew members’ experiences of empathy within and outside their in-groups to understand its impact on various performance measures and the overall group environment.

#### 1.1.3. Team Performance

Team performance pertains to the results obtained using multiple resources in pursuit of a team goal [61].

Similar to in other countries, full-service carriers in South Korea manage their cabin crew in teams/group units. Team performance is scored by combining the individual performances of cabin crew members, and the scores are compared with other teams to obtain a team unit score. Then, these results are reflected in the performance evaluation of individual team members [62].

Teamwork is extremely important for an airline cabin crew. For example, a team with a good bond between individual team members is better at achieving its goals, which is also reflected in the team’s performance levels [63].

Numerous studies have reported the positive effect of empathy toward colleagues on team performance. If the leader is empathetic, members can easily overcome failures and become motivated, which will have a positive effect on the overall performance of the organization [16].

Several studies have reported that empathy among colleagues has a positive impact on team/group performance [17]. Moreover, airline cabin crew with high empathic ability show stronger attachment to their working group, stronger organizational involvement, and better cooperation between colleagues; empathic ability also reduces workplace stress and is closely related to psychological comfort [18]. Therefore, we proposed the following hypothesis:

**Hypothesis** **6** **(H6):**
*Cabin crew with high empathy among colleagues show better team performance.*


#### 1.1.4. Organizational Atmosphere

Organizational atmosphere refers to the atmosphere formed by interpersonal relationships, behavior of group members, and other environmental variables [64], and it is also connected to group performance. For an airline cabin crew, the extent of emotional labor exerted by the members can be determined by its organizational atmosphere, wherein conversations with colleagues can relieve tension and strengthen bonds. Organizational atmosphere can also raise the morale of cabin crew and improve the quality of service [65,66].

Various studies on the South Korean domestic airline industry have demonstrated that the organizational atmosphere among cabin crew affects emotional labor, professionalism [67], and turnover intention [66], and that the organizational atmosphere within cabin crew teams affects organizational involvement [68]. In a study conducted on the causal relationship between empathy and organizational atmosphere, one study reported that empathy among organization members increases work satisfaction and lowers turnover intention [19].

Empathic ability has also been shown to reduce occupational stress among service workers, which improves the organizational atmosphere [20]. Moreover, worker empathy increases their job engagement and has positive effects on organizations [21].

In this study, we aimed to investigate whether rapport and empathy among airline cabin crew were also related to the behavioral variables of the organizational atmosphere. Therefore, we proposed the following hypothesis:

**Hypothesis** **7** **(H7):**
*Cabin crew members who show high empathy toward colleagues positively impact organizational atmosphere.*


#### 1.1.5. Irregularity

In organizations, accidents occur due to a combination of factors, and the number of accidents can be reduced by controlling or eliminating its contributing factors [22]. In the present study, we investigated irregularities resulting from human factors that lie outside the environmental factors. In the airline industry, human factors that cause irregularities pertain to mistakes by the cabin crew, cockpit crew, maintenance staff, ground crew, and air traffic controllers [69].

Human error by airline staff contributes to over 70% of aircraft accidents worldwide [70]. To prevent these accidents, several airlines have developed crew resource management (CRM) training programs as part of their safety education training for airline workers, which are used in airlines across the globe [71].

Irregularities resulting from human errors can manifest as mistakes or carelessness by the cabin crew, inaccurate situational awareness, tension within the work atmosphere, poor working conditions due to work stress, insufficient partnership and emotional friction among colleagues, insufficient communication and lack of information-sharing and feedback between the cabin crew, as well as delays in service due to teamwork failure [72].

Studies on the effects of empathy and irregularity have reported that the perceived fatigue within an airline cabin crew affects irregularity [23] and that communication levels within an organization, empathy, and a safe atmosphere reduced the rate of industrial accidents by safety and public health administrators [24]. In addition, during cabin crew briefing, the better the information that is communicated or the more focused the voice, the more effective it is in preventing accidents [22].

Based on these studies, we consider empathy toward colleagues as a factor that prevents irregularities within an organization. Therefore, we proposed the following hypothesis:

**Hypothesis** **8** **(H8):**
*A cabin crew with members who show greater empathy toward colleagues are more effective in preventing irregularities and can help airlines take retroactive measures for long-term safety.*


#### 1.1.6. Moderating the Effects of a Cabin Crew’s Rapport-Building Behavior within and Outside Their In-Group

Airlines manage and supervise cabin crew using a team-based system that places a high emphasis on teamwork and efficient human management.

Since 1997, Asiana Airlines follows a group management system, where each group consists of one group leader and 11–13 cabin crew members. Asiana Airlines comprises of three teams [73].

Korean Air has implemented a team system since 1986, with 390 groups of cabin crew, where each team is composed of a team leader, team vice-leader, and 13–15 team members on average, and the teams are maintained for one year at a time [74].

Team members inevitably improve their feelings of togetherness, intimacy, and bonding as they spend more time with other members while working on flights or away from work. However, there has been a shortage of research on rapport-building behaviors or empathy among the members of a cabin crew.

A multi-group analysis was performed to test whether there were any significant differences in the effects of rapport-building behaviors on empathy toward colleagues, which was moderated by whether the participants’ closest colleagues worked in the same team.

The path coefficients and their significance were investigated even though the paths from rapport-building behaviors to empathy toward colleagues showed no difference for the moderating role of the presence of the participants’ closest colleagues within the same team.

The results show that the effects of the four rapport-building behaviors were relatively evenly distributed when the participants’ closest colleagues were in the same team, whereas uncommonly attentive behavior had a stronger effect than other types of rapport-building behaviors in the case that the participants’ closest colleagues were not in the same team. Figure 1 provides the potential study model of this research. Therefore, we proposed the following hypotheses:

**Hypothesis** **9** **(H9):**
*The effects of uncommonly attentive behavior on empathy building among colleagues would be different for crew members within or outside one’s team.*


**Hypothesis** **10** **(H10):**
*The effects of common-grounding behavior on empathy building among colleagues would be different for crew members within or outside one’s team.*


**Hypothesis** **11** **(H11):**
*The effects of courteous behavior on empathy building among colleagues would be different for crew members within or outside one’s team.*


**Hypothesis** **12** **(H12):**
*The effects of connecting behavior on empathy building among colleagues would be different for crew members within or outside one’s team.*


**Hypothesis** **13** **(H13):**
*The effects of information-sharing behavior on empathy building among colleagues would be different for crew members within or outside one’s team.*


## 2. Materials and Methods

### 2.1. Study Design and Participants

In this study, we used a self-report questionnaire survey and convenience sampling to obtain responses from domestic and overseas airline cabin crew (Korean Air, 81 responses; Asiana Airlines, 99 responses; Air Busan, 2 responses; Air Seoul, 13 responses; Jeju Air, 16 responses; Jin Air, seven responses; T’way Airlines, one response; Fly Gangwon, four responses; Air China, seven responses; Etihad Airlines, one response).

The inclusion criteria were as follows: (1) cabin crew members currently working for an airline and (2) with at least one year of experience working in team/group flights.

During data collection, all cabin crew members who participated in the survey were informed that the collected information would remain private and would be destroyed after completing the analysis. After the participants gave their consent, they were provided with a link to an online survey via social networking sites (SNS) or emails. In addition, we administered face-to-face questionnaires within a cabin crew briefing room or through individual meetings.

We administered a total of 232 questionnaires through both online and face-to-face methods between June 1 and October 1, 2020. Of these, we excluded two responses that seemed insincere and included the remaining 230 questionnaires in the analysis.

### 2.2. Measures

To empirically measure the nine theoretical concepts proposed in this study, we used the following measures, which have been validated by previous studies from various fields such as cabin crew competency, communication, and psychology.

To measure the five dimensions of rapport-building behavior by cabin crew members, we used an interval scale consisting of three questions on uncommonly attentive behavior, two questions on common-grounding behavior, three questions on courteous behavior, two questions on connecting behavior, and three questions on information-sharing behavior, based on previous studies by Gremler and Gwinner [7] and Lee and Hyun [9].Empathy toward colleagues was measured using four questions selected from the Interpersonal Reactivity Index by Kim [58] and Davis [75].Team performance was measured using four questions selected from studies by Chiang [76], Lee, Nam, and Yang [77], and Kim and Cho [78].Organizational atmosphere was measured using the five questions proposed by Lee [67].Irregularity was measured using the three questions proposed by Oh [22].

After creating the initial questionnaire based on the above-described measures, we asked the questionnaire participants to respond to each question on a 5-point Likert scale ranging from “strongly disagree” (1 point) to “strongly agree” (5 points). To ensure the validity of the measures used in this study, we conducted a preliminary interview survey with a focus group consisting of cabin crew members from full-service carriers in South Korea before the administering the questionnaire. Next, we conducted a pilot test on 30 members from a domestic cabin crew to check the readability of the questionnaire. Based on the preliminary survey, we made several rounds of improvements and adjustments, and we administered the questionnaire after a final check by a professor specializing in the subject. We obtained a Cronbach’s α of higher than 0.7, suggesting that the scales used in this study were reliable.

### 2.3. Data Analysis

After receiving the survey responses, we performed the following steps. First, we performed a frequency analysis to ascertain the participants’ general characteristics. Second, we performed a confirmatory factor analysis (CFA) to evaluate the validity of the measured model and tested the convergent and discriminant validities. Cronbach’s α was calculated to evaluate the reliability of the scales used. Third, we performed structural equation modeling (SEM) to analyze the relationships between the variables. Fourth, a bootstrap analysis was performed to test the indirect effects of rapport-building behavior on team/group performance, organizational atmosphere, and response to irregularities, which were mediated by empathy among crew members. Fifth, we conducted a multi-group analysis to test whether the relationships between the variables showed significant differences, depending on whether the participant’s closest colleague was on the same team.

We used IBM SPSS 25 and AMOS 25 to conduct our statistical analyses, and determined its statistical significance at a 5% significance level.

## 3. Results

### 3.1. Demographic Profile of the Respondents

To achieve the objectives of our study, we collected a sample of 230 individuals of which 19 were male (8.3%) and 211 were female (91.7%). By age, we had 4 participants (1.7%) aged 20–25 years. 47 participants (20.4%) aged 26–30 years, 110 participants (47.8%) aged 31–35 years, 52 participants (22.6%) aged 36–40 years, 14 participants (6.1%) aged 41–45 years, and 3 participants (1.3%) aged above 45 years. Our sample contained 118 unmarried participants (51.3%) and 112 married participants (48.7%). During the survey period, 32 participants (13.9%) had graduated from a professional college, 162 participants (70.4%) had graduated from college, 21 participants (9.1%) were currently enrolled in graduate school, and 15 participants (6.5%) had graduated from graduate school or above. Additionally, 12 participants (5.2%) had less than 2 years of experience, 40 participants (17.4%) had 2–5 years of experience; 103 participants (44.8%) had 6–10 years of experience; 54 participants (23.5%) had 11–15 years of experience; and 21 participants (9.1%) had more than 15 years of experience. Regarding the participants’ job titles, 115 participants (50.0%) were stewards/stewardesses, 83 participants (26.1%) were assistant pursers, 29 participants (12.6%) were pursers, and 3 participants (1.3%) were senior pursers. Salary wise, 16 participants (7.0%) received an annual salary of less than 30 million KRW, 73 participants (31.7%) received 30–40 million KRW; 71 participants received 41–50 million KRW; 37 participants (16.1%) received 51–60 million KRW, and 33 (14.3%) participants received a salary of more than 60 million per annum.

After eliminating instances of double counting, 50% of the participants reported being on the same team as their closest colleague (115 persons).

### 3.2. Results of Confirmatory Factor Analysis and Reliability Analysis

A CFA of the rapport-building behaviors revealed that the correlation coefficient between “uncommonly attentive behavior” and “common-grounding behavior” had a value close to 0.9. The fact that such diverse latent variables showed such a strong correlation harmed the discriminant validity of the model. Additionally, the main survey of the cabin crew members also showed a high similarity between the two factors. Therefore, we combined the five questions regarding “uncommonly attentive behavior” and “common-grounding behavior” into a single factor and reperformed the CFA.

Table 1 shows the results from the CFA; all questions showed a high factor loading of over 0.50. The model showed CFI = 0.940, Tucker–Lewis Index (TLI) = 0.930, and root mean square error of approximation (RMSEA) = 0.054, which satisfied the validity criteria and demonstrated a good fit.

The internal consistency of the questions in each factor was analyzed using Cronbach’s α, and all factors showed a coefficient greater than 0.70, demonstrating that the instrument was reliable.

### 3.3. Convergent Validity Analysis

Table 2 presents the results of the variables’ composite reliability (CR) and average variance extracted (AVE), which were used to analyze their convergent validity. All variables showed CR greater than 0.70 and AVE greater than 0.50, suggesting a satisfactory convergent validity.

### 3.4. Discriminant Validity Testing

All pairs of the latent variables in our model showed positive correlation coefficients; therefore, we tested their discriminant validity after verifying that the upper bounds at the 95% confidence intervals did not exceed 1. The results are shown in Table 3. The correlation coefficients are shown below the diagonal in the table, and the upper bounds at the 95% confidence intervals are shown above the diagonal. None of the upper bounds exceeded 1. This implies that the correlation coefficients between the pairs of latent variables in our study were not too high, and the instrument showed a satisfactory discriminant validity.

The results showed a strong correlation (0.891) between organizational atmosphere and irregularity among cabin crew members; therefore, we additionally performed a discriminant validity test proposed by Bagozzi and Yi. To this end, we compared the chi-squared values of a model merging organizational atmosphere and irregularity into a single factor with the original model where these two factors were separate. The results are listed in Table 4.

The chi-squared value of the original model was 582.366 with 349 degrees of freedom. Meanwhile, the chi-squared value of the merged model was 606.704, with 356 degrees of freedom. Hence, the chi-squared values had a difference of 24.388 and the degrees of freedom had a difference of 7; however, the critical chi-squared value for 7 degrees of freedom was 14.067. As the difference between the two models was greater than the critical value, the model in which organizational atmosphere and irregularity were separate was the superior model.

Overall, we determined that organizational atmosphere and irregularity showed a discriminant validity.

### 3.5. Descriptive Statistics and Normality Testing

Table 5 shows the descriptive statistics of the major variables used in this study. Among the various types of rapport-building behaviors, the mean score of uncommonly attentive behavior was 4.14, courteous behavior was 4.22, connecting behavior was 4.08; and information-sharing behavior was 4.01. The mean score for the mediating variable, empathy toward colleagues, was 4.09. For the dependent variables—team performance, organizational atmosphere, and irregularity among crew members, the mean scores were 3.67, 4.00, and 3.87, respectively, indicating that scores were overall above average.

We also analyzed the skewness and kurtosis of the results to determine whether they satisfied the assumption of normality. The results were below the criterion values for all variables, indicating that the data satisfied the assumption of normality.

### 3.6. SEM and Goodness of Fit

We investigated the effects of rapport-building behaviors on empathy toward colleagues, and the effects of empathy toward colleagues on team performance, organizational atmosphere, and irregularities. We also measured the moderating effect of empathy toward colleagues on team performance, organizational atmosphere, and irregularities to measure the indirect effect of rapport-building behaviors on these parameters. The variables “uncommonly attentive behavior” and “common-grounding behavior”, which were merged because of issues with discriminant validity in the original model, were treated as a single factor for SEM.

We used the following indices of fit to analyze the fit of the SEM constructed in this study; the results are shown in Table 6.

The main indices of fit were CFI = 0.939, TLI = 0.931, and RMSEA = 0.054. Since CFI and TLI were both greater than 0.90, and RMSEA was lower than 0.08, the model was determined as showing a good fit.

### 3.7. Testing Direct Effects of Rapport-Building Behaviors

To test the direct effects between the variables in the SEM, we analyzed the significance of the path coefficients (Table 7).

First, we tested the paths from rapport-building behaviors to crew members’ empathy toward their colleagues, and we found significant positive paths from uncommonly attentive behavior to empathy toward colleagues (β = 0.368, *p* < 0.001), from courteous behavior to empathy toward colleagues (β = 0.221, *p* < 0.05), from connecting behavior to empathy toward colleagues (β = 0.229, *p* < 0.01), and from information-sharing behavior to empathy toward colleagues (β = 0.193, *p* < 0.05). This implied that the higher the scores for each component of rapport-building behavior, the stronger the empathy exhibited by crew members toward their colleagues. Upon comparing the standardized coefficients, we found that uncommonly attentive behavior (β = 0.368) had the strongest effect on empathy toward colleagues, followed by—in descending order—connecting behavior (β = 0.229), courteous behavior (β = 0.221), and information-sharing behavior (β = 0.193).

Next, we analyzed the effects of cabin crew members’ empathy toward colleagues on the dependent variables, and found significant positive paths, from empathy toward colleagues to team performance (β = 0.363, *p* < 0.001), organizational atmosphere (β = 0.432, *p* < 0.001), and irregularity scores (β = 0.417, *p* < 0.001). This demonstrated that the stronger the empathy toward one’s colleague, the better the cabin crew’s team performance, organizational atmosphere, and irregularity scores. After comparing the standardized coefficients, we found that the effect of empathy toward colleagues was the strongest for organizational atmosphere (β = 0.432), followed by irregularity scores (β = 0.417) and team performance (β = 0.363). Figure 2 describes the hypothesis testing results.

### 3.8. Testing Indirect Effects of Rapport-Building Behaviors

We performed a bootstrap analysis to test the indirect effects of rapport-building behaviors on cabin crew’s team performance, organizational atmosphere, and irregularity, which are mediated by empathy toward colleagues (results in Table 8). We selected 2000 bootstrap samples and determined the statistical significance based on the 95% confidence interval.

Uncommonly attentive behavior (β = 0.133, *p* < 0.01), courteous behavior (β = 0.080, *p* < 0.05), connecting behavior (β = 0.083, *p* < 0.01), and information-sharing behavior (β = 0.070, *p* < 0.05) all had significant indirect effects on team performance, with empathy toward colleagues as a mediator; meanwhile, uncommonly attentive behavior (β = 0.159, *p* < 0.01), courteous behavior (β = 0.095, *p* < 0.05), connecting behavior (β = 0.099, *p* < 0.01), and information-sharing behavior (β = 0.083, *p* < 0.05) all had significant indirect effects on organizational atmosphere, with empathy toward colleagues as a mediator; lastly, uncommonly attentive behavior (β = 0.153, *p* < 0.01), courteous behavior (β = 0.092, *p* < 0.05), connecting behavior (β = 0.095, *p* < 0.01), and information-sharing behavior (β = 0.080, *p* < 0.05) all had significant indirect effects on irregularity.

In summary, empathy toward colleagues mediated the positive effects of all the rapport-building behaviors on team performance, organizational atmosphere, and irregularities.

### 3.9. Testing the Moderating Effect of the Participants’ Closest Colleagues Being in the Same Team

We performed a multi-group analysis to test whether there were any significant differences in the effects of rapport-building behaviors on empathy toward colleagues moderated by whether the participants’ closest colleagues worked in the same team.

We compared a restricted model, in which the paths from rapport-building behaviors to empathy toward colleagues were identically restricted, depending on whether or not the participant’s closest colleagues worked in the same team, against a model with no restrictions (Table 9). The difference in the chi-squared statistics between the restricted and the unrestricted models was 0.928; however, it did not exceed the critical value of 9.488 when the difference in the degrees of freedom was 4. This implies that the two models did not exhibit any significant difference.

However, even though the paths from rapport-building behaviors to empathy toward colleagues showed no difference for the moderating role of the presence of the participants’ closest colleagues within the same team, we investigated the path coefficients and their significance (Table 10).

The results showed that when the participants’ closest colleagues were in the same team, the effects of the four rapport-building behaviors were relatively evenly distributed. Meanwhile, when the participants’ closest colleagues were not in the same team, uncommonly attentive behavior had a stronger effect than other types of rapport-building behaviors.

### 3.10. Hypothesis Testing

Our SEM analysis revealed that all four types of rapport-building behaviors: uncommonly attentive behavior, courteous behavior, connecting behavior, and information-sharing behavior had significant positive effects on the crew members’ empathy toward their colleagues, which in turn had significant positive effects on team performance, organizational atmosphere, and the crew members’ irregularity scores. Contrastingly, we found no significant moderating effect of having one’s closest colleague in the same team. The results of the SEM analysis are summarized in Table 11.

## 4. Discussion

South Korea’s airline policies do not contain a manual for cabin crew members to regulate unreasonable demands from customers. Airline workers perform immense emotional labor in a high-intensity work environment. However, there is a dearth of accurate measures or methods to conduct an in-depth verification of whether airlines provide their cabin crew with a safe workplace environment that makes them feel protected. There is also a limited body of research on strategies through which workers can manage themselves, and both corporate and social measures to solve these problems are either absent or insufficient.

Therefore, we believe that there is an urgent need to develop measures to safeguard cabin crew members from coercion at the workplace and to protect their human rights at the company, society, and trans-governmental levels. The present study was part of an effort to alter consumers’ and companies’ perceptions of airline cabin crew working both domestically and internationally. In this study, we focused on rapport-building behaviors and feelings of empathy among cabin crew members as a means to improve their working conditions through mutual support and communication.

Our study showed that the impact of rapport-building behaviors on cabin crew members’ empathy toward colleagues could be seen in the following order from the strongest to the weakest: (1) uncommonly attentive behavior; (2) connecting behavior; (3) courteous behavior; and (4) information-sharing behavior.

### 4.1. Analysis of the Results

We interpreted the results through the following aspects. First, cabin crew members who frequently performed uncommonly attentive behaviors were found to have a stronger ability to empathize with their colleagues. Therefore, initiating new conversations, showing a strong level of interest, and attentively taking care of colleagues are some behaviors that can build empathy among colleagues. These findings are consistent with a previous study, which shows that among the five types of rapport-building behaviors, uncommonly attentive behavior had the strongest effect on positive emotions among tour guides [10]. Thus, from the perspective of human resource management in the airline industry, giving or receiving interest or attentive care from colleagues increases empathy and has positive effects on cabin crew members’ psychological health, work life, and overall quality of life.

Second, cabin crew members who frequently performed courteous behaviors had a stronger ability to empathize with their colleagues, which can be exemplified by being respectful toward one’s colleagues and apologizing during tense situations. This is consistent with a previous study that showed that good manners increased employees’ rapport with their customers [7].

Third, strong empathic behavior shown by cabin crew members who frequently performed connecting behaviors could be because of their efforts to build connections with fellow members through humor and a pleasant environment. A previous study on connecting behaviors between tour guides and Chinese tourists did not show a statistically significant relationship with cognitive or emotional empathy. However, in this study, we found a significant correlation between connecting behaviors and empathy toward colleagues in the context of airline crew [9]. The staff from airline cabin crew are constantly shifted to new assignments and frequently encounter new aircrafts, routes, and colleagues. Therefore, the ability to rapidly resolve a tense atmosphere is crucial while working on flights.

Fourth, cabin crew members who showed a strong sense of empathy toward their colleagues frequently performed information-sharing behaviors such as making useful proposals about flight work and sharing one’s expert knowledge with newer members. This is consistent with a previous study on hotel service workers and customers where information-sharing had significant effects on emotional well-being [40].

However, among the various rapport-building behaviors, information-sharing behavior showed the weakest effect on empathy toward colleagues. As professionals, cabin crew members strive to provide useful information and advice to their colleagues during work. However, its relatively weak impact could be explained by the participants’ tendency to think of this as an obvious duty that is part of their jobs.

Fifth, cabin crew members with a strong ability to empathize with their colleagues corresponded with an improved team performance. Empathy among cabin crew members makes them ensure that their team provides the best service while effectively meeting work targets. This is consistent with a previous study, wherein a leader’s empathy was perceived as a positive factor by the staff, which also reflected positively on their work performance [16]. Therefore, airlines should promote a system that enables airline crew members to visit each other freely at any time and receive help for their personal stress management or psychological/emotional care. In the long term, improving mechanisms to facilitate communication within and between the cabin crew members would also benefit companies by helping to improve their performance levels. Some solutions include providing staff with access to independent psychological counseling centers with no risk to the security of their personal information, support for psychological education, or opportunities for rest time and sufficient holidays away from high-intensity work.

Sixth, empathic cabin crew members positively affected the organizational atmosphere. This could stem from the staffs’ tendency to help colleagues perform difficult tasks and make them feel comfortable while working on the same flight. This result conforms with a previous study showing that a positive organizational atmosphere had a positive impact on employees’ work attitudes [79] and showed that an organization’s positive atmosphere increased workers’ morale, facilitated self-realization, and enhanced organizational productivity. Modern industrial societies, including the airline and tourism industry, heavily rely on human resources. Therefore, it is extremely important for companies to effectively manage human resources and allow workers to fully realize their capabilities. For this, they need to place high importance on aspects of strategic management, such as strengthening workers’ qualities and abilities and improving communication and teamwork. Efficient human resource management would inevitably result in a more participatory organizational atmosphere, which will in turn lead to better business outcomes.

Seventh, empathic cabin crew members were found to be better equipped to rapidly prevent and respond to irregularities. Specifically, cabin crew members who were good at empathizing with their colleagues showed fewer mistakes during flight work and experienced higher work satisfaction after completing a shift without making mistakes. This is consistent with a previous study in which administrators’ empathy and a safe atmosphere helped reduce the rate of industrial accidents [24]. Similarly, we found that the facilitation of good communication by empathic cabin crew members could provide the foundations for safe flight work by reducing mistakes and irregularities due to human factors, thus enhancing the whole team’s preventive safety. This provides enormous value through positive effects on airline safety assessments, because it can prevent accidents not only for the customers onboard but also for the cockpit and cabin crew.

Eighth, our hypothesis on the variation in the effects of rapport-building behaviors on empathy toward colleagues between cabin crew members in the same, or different teams, was rejected. The paths between rapport-building behaviors and empathy did not show any clear differences depending on whether the participant’s closest colleague was on the same team.

The following sections discuss the implications of these results.

### 4.2. Implications

#### 4.2.1. Academic Implications

First, previous studies on rapport-building behaviors include a study investigating the relationship between restaurant workers and customers [8], a study on rapport-building behaviors and empathy between tour guides and Chinese tour groups [9], and rapport-building behaviors among hotel service workers [40]. Thus, to date, studies on the effect of rapport-building behaviors have been limited to certain relationships between workers and clients. In the present study, we contributed to the theoretical literature by expanding the scope of participants to rapport-building behavior between airline cabin crew rather than toward customers.

Second, numerous studies have aimed at improving the performance and company services offered by cabin crew to their customers. However, few studies have considered rapport-building behaviors and empathy between airline cabin crew members. To the best of our knowledge, this was the first such study to investigate these factors among airline cabin crew colleagues. This study also provides academic value by identifying which of the five types of rapport-building behavior plays the biggest role in forming empathy among colleagues.

#### 4.2.2. Practical Implications

First, by deducing the various rapport-building behaviors exhibited by airline cabin crew, this study’s results have implications for airline companies’ strategies to improve the human resource management of airline cabin crew members, and the study also contributes suggestions to improve the quality of the work environment and corporate culture among the cabin crew. It also helps to improve interpersonal relationships between the existing airline staff.

Second, these results suggest that creating better working environments to form good relationships between cabin crew members is essential to reduce negative human factors due to workplace stress resulting from conflict among colleagues.

Third, the findings suggest that rather than relying on the individual efforts of the staff, the airline industry should support projects at the national level, such as providing programs to improve mental health and quality of life to protect not just airline workers, but all workers in the tourism industry from excessive emotional labor.

Fourth, the effects of rapport-building behaviors demonstrated in this study would also lead to higher user satisfaction from airlines’ services and products, including cabin crew competency.

## 5. Conclusions

For an airline cabin crew, communication and empathy between colleagues are extremely important for a congenial work environment. Team members frequently interact with each other from the briefing stage before starting the flight to carrying out their work during the flight. During regular safety training, including emergency water landing drills, the cabin crew members need to have strong empathy and be attuned to each other’s thoughts, so that they can cooperate and prevent emergency situations or making excessive demands on each other, allowing them to respond rapidly with minimal loss. In this regard, rapport-building behaviors and empathy are extremely important, even beyond the basic demands of cordial relationships between colleagues.

Cabin crew work in a team-based system and constantly meet new colleagues due to varied work schedules. The results of this study indicated that rapport-building behaviors between colleagues and empathy toward colleagues increased team performance, improved the organizational atmosphere, reduced the rate of irregularities, and also had positive effects on timely responses to irregularities through active communication and cooperation between cabin crew colleagues. Our study on rapport-building behaviors and empathy among cabin crew members serves as a basis for research on a broader population. By demonstrating an important factor for the reduction of safety-related accidents (irregularities) that are directly related to organizational and team performance and human lives, this research highlighted the influence and scope of research on airline cabin crew.

### Limitation and Recommendation for Future Research

This study has several limitations, based on which we suggest some directions for future research to address these limitations.

First, during the study’s survey period, the airline industry was in the midst of a serious management crisis due to the COVID-19 pandemic, which at the time of the research was threatening the very existence of the airline industry. During the survey, many cabin crew members were on an unpaid or rotational leave. This detachment from the direct experience of the working environment could most certainly have affected the results of the survey. A comparative study conducted by repeating the survey after the airline industry has recovered and the cabin crew returned to normal work schedules would provide better insights on the accuracy of the results.

Second, the moderating effect of the presence of the participants’ closest colleagues within the same team on the relationship between rapport-building behaviors and empathy among cabin crew colleagues was rejected. The COVID-19 pandemic has led to a sustained decrease in demand for the airline industry, and the cutbacks in the number of flights and personnel have led to an increase in the number of cabin crew on leave. Currently, among the domestic full-service carriers, Asiana Airlines has been acquired by Korean Air, which plan to merge soon. Due to staffing reductions, there has been a serious lack of flight work for the airline crew. Thus, the shortage of flights to investigate the differences between cabin crew members on the same/different teams meant that the moderating variable was not fully represented by the measured survey items.

Third, the sample mostly consisted of cabin crew working for domestic full-service carriers (Korean Air, Asiana Airlines). Therefore, future studies could expand the framework of this study to include cabin crew members from diverse low-cost carriers and overseas airlines to improve the generalizability of the results.

Fourth, a broader and larger sample could not be obtained because of social distancing and other environmental constraints, given that the survey was conducted after the onset of the COVID-19 pandemic. Thus, it is difficult to generalize the research results owing to the small sample size. It is imperative to collect more samples for greater result reliability of future studies.

Fifth, the notion of rapport has been studied in the consumers behavior field. This study expanded the existing study into the human resource area. For the future study, it is necessary to link the rapport with the mainstream of human resource areas, such as leader–member exchange, employee work stress, turnover rate, and employee well-being.

## Figures and Tables

**Figure 1 ijerph-18-06417-f001:**
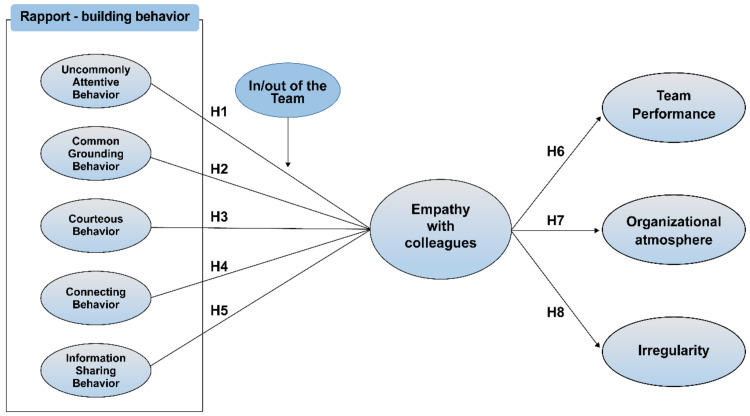
Conceptual model.

**Figure 2 ijerph-18-06417-f002:**
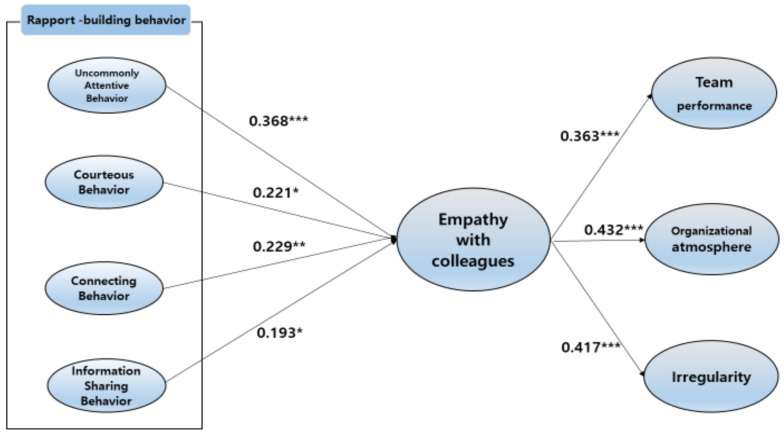
Results from the SEM model. (* *p* < 0.05, ** *p* < 0.01, *** *p* < 0.001.)

**Table 1 ijerph-18-06417-t001:** Results of CFA and reliability analysis.

Factor	Question	M(SD)	Loading	Cronbach’s α
Uncommonly attentive behavior	My colleagues take special care of me during and outside of working hours.	4.13(0.79)	0.726	0.861
	I have been emotionally moved by my colleagues’ gestures.	4.31(0.81)	0.741	
	My colleagues show a strong interest in me beyond concerns about work.	3.93(0.88)	0.745	
	My colleagues initiate or try to sustain conversations with me and make an effort to find common interests.	4.16(0.71)	0.805	
	Apart from shared interests, my colleagues make an effort to perform activities that we can enjoy together.	4.17(0.78)	0.715	
Courteous behavior	My colleagues apologize sincerely for their mistakes when a problem arises during service.	4.11(0.78)	0.804	0.866
	My colleagues are pleasant, polite, and helpful, and they always behave respectfully.	4.25(0.73)	0.797	
	My colleagues are friendly toward me and act in a way that makes me feel empathic. They show concern and apologize when a problem arises.	4.30(0.71)	0.878	
Connecting behavior	My colleagues make an effort to connect with me through humor.	4.04(0.79)	0.743	0.801
	My colleagues always make an effort to create an enjoyable atmosphere.	4.12(0.77)	0.900	
Information-sharing behavior	My colleagues provide useful suggestions regarding flight-related work.	3.87(0.90)	0.702	0.857
	My colleagues share their professional knowledge about flight work with me.	4.12(0.85)	0.868	
	My colleagues share professional knowledge with me so I may enjoy a safe flight and provide smooth service.	4.05(0.89)	0.894	
Empathy toward colleagues	I consider other crew members more than my colleagues; I have empathy and understanding for them just like I have for my family. They also talk to me about their personal lives.	4.08(0.89)	0.702	0.768
	Before doing anything, I always try to think about how my colleagues would feel.	3.98(0.82)	0.642	
	I can easily anticipate my colleagues’ thoughts and reactions.	4.06(0.68)	0.621	
	Sometimes, I imagine things from my colleagues’ perspective and make an effort to understand them better.	4.22(0.69)	0.758	
Team/group performance	I think that our team performs better than other teams.	3.65(0.87)	0.837	0.902
	My team provides better quality of service than other teams.	3.76(0.82)	0.882	
	My team receives more praise from customers than other teams.	3.56(0.85)	0.779	
	My team achieves work goals more effectively than other teams.	3.71(0.84)	0.846	
Organizational atmosphere	My team members share good/friendly relationships with each other.	4.04(0.81)	0.827	0.900
	My team members help each other when performing difficult tasks.	4.11(0.79)	0.857	
	My team members feel comfortable when working flights together.	4.14(0.89)	0.775	
	My team members show active intention to participate in social gatherings.	3.73(1.02)	0.747	
	My team members can share good news with each other.	4.00(0.90)	0.826	
Irregularity	My team shows fewer irregularities than other teams.	3.70(0.87)	0.529	0.749
	Compared to other teams, the cabin crew in my team make fewer major mistakes.	3.89(0.81)	0.662	
	Compared to other teams, my team shows much higher work satisfaction after completing a flight.	4.03(0.77)	0.852	

SD = standard deviation; χ2 = 582.366 (df = 349, *p* < 0.001); CFI = 0.940; Tucker–Lewis Index (TLI) = 0.930; root mean square error of approximation (RMSEA) = 0.054.

**Table 2 ijerph-18-06417-t002:** Convergent validity testing.

Variable	Composite Reliability (CR)	Average Variance Extracted (AVE)
Uncommonly attentive behavior	0.908	0.665
Courteous behavior	0.921	0.796
Connecting behavior	0.872	0.775
Information-sharing behavior	0.890	0.732
Empathy toward colleagues	0.853	0.593
Team performance	0.928	0.764
Organizational atmosphere	0.921	0.700
Irregularity	0.795	0.573

**Table 3 ijerph-18-06417-t003:** Discriminant validity testing based on correlation coefficient confidence intervals.

Variable	1	2	3	4	5	6	7	8
1. Uncommonly attentive behavior	1	(0.757)	(0.575)	(0.656)	(0.825)	(0.305)	(0.403)	(0.340)
2. Courteous behavior	0.684	1	(0.568)	(0.719)	(0.773)	(0.355)	(0.476)	(0.445)
3. Connecting behavior	0.508	0.497	1	(0.461)	(0.675)	(0.316)	(0.326)	(0.352)
4. Information-sharing behavior	0.585	0.641	0.394	1	(0.696)	(0.444)	(0.477)	(0.427)
5. Empathy toward colleagues	0.745	0.691	0.595	0.614	1	(0.411)	(0.454)	(0.433)
6. Team performance	0.240	0.284	0.247	0.370	0.335	1	(0.750)	(0.774)
7. Organizational atmosphere	0.342	0.407	0.263	0.406	0.383	0.662	1	(0.977)
8. Irregularity	0.293	0.390	0.303	0.372	0.376	0.694	0.891	1

Below the diagonal: correlation coefficients; above the diagonal: upper bounds at the 95% confidence intervals for the correlation coefficients.

**Table 4 ijerph-18-06417-t004:** Discriminant validity testing using the method proposed by Bagozzi and Yi.

Model	χ^2^	df	Δχ2	Δdf	CFI	TLI	RMSEA
Original Model	582.366	349	24.338	7	0.940	0.930	0.054
Merged Model	606.704	356			0.936	0.927	0.055

χ^2^ = chi-squared value; df = degree of freedom; **Δ**χ^2^ = difference in chi-squared value; Δdf = difference in degree of freedom.

**Table 5 ijerph-18-06417-t005:** Descriptive statistics of the major variables.

Variable	Possible Range	Mean	Standard Deviation	Skewness	Kurtosis
Uncommonly attentive behavior	1–5	4.14	0.64	−0.61	0.21
Courteous behavior	1–5	4.22	0.66	−0.63	0.09
Connecting behavior	1–5	4.08	0.72	−0.59	−0.01
Information-sharing behavior	1–5	4.01	0.78	−0.57	0.11
Empathy toward colleagues	1–5	4.09	0.60	−0.30	−0.19
Team performance	1–5	3.67	0.75	0.04	−0.57
Organizational atmosphere	1–5	4.00	0.75	−0.51	−0.38
Irregularity	1–5	3.87	0.67	−0.09	−0.06

**Table 6 ijerph-18-06417-t006:** Structural equation model fit.

χ^2^	df	*p*	CFI	TLI	RMSEA
599.618	361	<0.010	0.939	0.931	0.054

**Table 7 ijerph-18-06417-t007:** Results of the SEM analysis.

Path			B	SE	β	C.R.	*p*
Uncommonly attentive behavior	→	Empathy toward colleagues	0.395	0.104	0.368	3.798 ***	<0.001
Courteous behavior	→	Empathy toward colleagues	0.215	0.095	0.221	2.268 *	0.023
Connecting behavior	→	Empathy toward colleagues	0.239	0.077	0.229	3.107 **	0.002
Information-sharing behavior	→	Empathy toward colleagues	0.187	0.080	0.193	2.353 *	0.019
Empathy toward colleagues	→	Team performance	0.429	0.092	0.363	4.666 ***	<0.001
Empathy toward colleagues	→	Organizational atmosphere	0.471	0.086	0.432	5.489 ***	<0.001
Empathy toward colleagues	→	Irregularity	0.312	0.070	0.417	4.449 ***	<0.001

* *p* < 0.05, ** *p* < 0.01, *** *p* < 0.001.

**Table 8 ijerph-18-06417-t008:** Results from testing indirect effects.

Path			β	SE	95% CI	*p*
Uncommonly attentive behavior	→	Team performance	0.133 **	0.047	0.052–0.237	0.006
Courteous behavior	→	Team performance	0.080 *	0.048	0.001–0.199	0.046
Connecting behavior	→	Team performance	0.083 **	0.033	0.028–0.161	0.003
Information-sharing behavior	→	Team performance	0.070 *	0.040	0.002–0.165	0.039
Uncommonly attentive behavior	→	Organizational atmosphere	0.159 **	0.063	0.046–0.299	0.007
Courteous behavior	→	Organizational atmosphere	0.095 *	0.054	0.004–0.223	0.039
Connecting behavior	→	Organizational atmosphere	0.099 **	0.037	0.038–0.182	0.002
Information-sharing behavior	→	Organizational atmosphere	0.083 *	0.047	0.004–0.192	0.037
Uncommonly attentive behavior	→	Irregularity	0.153 **	0.060	0.051–0.292	0.006
Courteous behavior	→	Irregularity	0.092 *	0.053	0.003–0.215	0.041
Connecting behavior	→	Irregularity	0.095 **	0.037	0.033–0.181	0.003
Information-sharing behavior	→	Irregularity	0.080 *	0.044	0.004–0.182	0.037

* *p* < 0.05, ** *p* < 0.01.

**Table 9 ijerph-18-06417-t009:** Testing the moderating effect of the participant’s closest colleagues being in the same team.

Model	χ^2^	df	Δχ2	Δdf	CFI	TLI	RMSEA
Unrestricted model	1040.357	722			0.918	0.908	0.044
Restricted model	1041.285	726	0.928	4	0.919	0.910	0.044

**Table 10 ijerph-18-06417-t010:** Effects of rapport-building behaviors on empathy toward colleagues moderated by the presence of participant’s closest colleague within the same team.

Path			Same Team (*n* = 115)		Different Teams (*n* = 115)	
			β	*p*	β	*p*
Uncommonly attentive behavior	→	Empathy toward colleagues	0.322 *	0.014	0.439 **	0.001
Courteous behavior	→	Empathy toward colleagues	0.202	0.084	0.258	0.139
Connecting behavior	→	Empathy toward colleagues	0.237 *	0.034	0.201 *	0.040
Information-sharing behavior	→	Empathy toward colleagues	0.248 *	0.037	0.111	0.389

* *p* < 0.05, ** *p* < 0.01.

**Table 11 ijerph-18-06417-t011:** Hypothesis testing.

	Path	Result	Standardized Coefficient (β)
1	Uncommonly attentive behavior → Empathy with colleagues	Supported	0.368
2	Courteous behavior → Empathy toward colleagues	Supported	0.221
3	Connecting behavior → Empathy toward colleagues	Supported	0.229
4	Information-sharing behavior → Empathy toward colleagues	Supported	0.193
5	Empathy toward colleagues → Team performance	Supported	0.363
6	Empathy toward colleagues → Organizational atmosphere	Supported	0.432
7	Empathy toward colleagues → Irregularity scores	Supported	0.417
8	Moderating effect of having one’s closest colleague in the same team	Rejected

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
