# Peer review of "Influence of Airline Cabin Crew Members’ Rapport-Building Behaviors and Empathy toward Colleagues on Team Performance, Organizational Atmosphere, and Irregularity"

_ijerph, 2021, doi:10.3390/ijerph18126417_

Round 1
Reviewer 1 Report
Overall, a very interesting research topic and investigation into understanding the effects of airline cabin crews members' rapport-building behaviors and empathy toward colleagues on team performance etc. I think this is a research area that needs to be further investigated. The purpose of the paper is clear and the methodology seems suitable. The manuscript reads fluently. However, there are several comments needed to be handled as well. Please, find my comments below:
- Introduction needs a major revision. Please, clearly state the purpose of study more clearly. You may need to state the gap existing in the literature. Based on your finding related to the gap, the objective of the study should be more clearly stated.
- In introduction, please provide implications of the current research -both theoretically and practically - and originality clearly.
- Theoretical evidence to support hypotheses are lack. Focus on the underlying mechanism for each hypothesis, and please provide them.
- H9 - H13 (moderating hypotheses) would be better if it is more specific (such as, focusing on how different?)
Author Response
Reviewer #1:
Overall, a very interesting research topic and investigation into understanding the effects of airline cabin crew members' rapport-building behaviors and empathy toward colleagues on team performance etc. I think this is a research area that needs to be further investigated. The purpose of the paper is clear and the methodology seems suitable. The manuscript reads fluently. However, there are several comments needed to be handled as well. Please, find my comments below.
Response
Thank you for your time and effort in reviewing our paper. We have thoroughly reviewed your comments and addressed them to the best of our abilities. Our responses to your comments are presented below.
- First, Introduction needs a major revision. Please, clearly state the purpose of study more clearly. You may need to state the gap existing in the literature. Based on your finding related to the gap, the objective of the study should be more clearly stated.
Response
In accordance with your suggestions, we have revised the purpose of the study to state it more clearly. In addition, we have revised the Introduction as shown below.
- Introduction
To date, in South Korea, there has been a constant interest regarding ways to improve airline cabin crews’ performance and the services offered to the customers. However, most recent studies have focused on ensuring the human rights of individual cabin crew members and on other psychological and emotional aspects of the job [1-3].
The purpose of this study is to confirm the effectiveness of rapport-building behavior among cabin crew members in enhancing their cooperation with each other and increasing the synergy effect achieved.
A literature review revealed that rapport-building behavior strengthens the psychological trust between the service provider and the customer, and additionally, improves the quality of interaction between them [4]. Furthermore, the communication between the leader and members, i.e., "leader–member" communication, is viewed as an extremely important factor. Therefore, a method that leaders can utilize to improve the quality of the relationship was presented, and a model of leadership communication was suggested based on the rapport management theory [5]. Additionally, there are studies that report that the more empathetic the leader is, the higher the member’s job satisfaction is likely to be [6]. Therefore, this study expanded on previous research and focused on the relationship among cabin crew members. However, few studies have focused on rapport-building behaviors and empathy among cabin crew colleagues. In this study, we considered aspects such as airline workers’ working conditions and dignity of labor and expanded the concepts of rapport and empathy from the context of basic human relationships to the context of airline cabin crew members.
Cabin crew members in Korea's domestic airlines work as a team. A few previous studies show that psychological bonds are important for rapport building, which is crucial for cabin crew members who always work in close cooperation. Therefore, this study also focused on rapport-building behaviors.
Based on a comprehensive literature review, we classified rapport-building behavior between cabin crew members into five categories: (1) uncommonly attentive behavior between the crew members, (2) common-grounding behavior, (3) courteous behavior, (4) connecting behavior, and (5) information-sharing behavior [7-10]. We then investigated the behaviors that had the strongest effect on empathy among the crew members.
Because a cabin crew works within a group/team, a psychological bond between colleagues is essential to help the members intuitively identify instances when their colleagues experience difficulties during their work and the type of assistance they may require. Academics have defined this type of psychological bond as empathy [11-13]. Empathy refers to a strong, deep psychological connection between two individuals, which can be formed through smooth communication (rapport building) [14,15]. In this study, it was found that rapport-building behavior has a positive effect on empathy.
Previous studies have reported that strong empathy with colleagues can have a positive effect on team performance [16-18], organizational atmosphere [19-21] while preventing irregularity among employees [22-24]. In this study, we investigated the causality of the relationship between empathy and team performance, organizational atmosphere, and employee irregularity. We also performed a statistical analysis of the mediating role of having one’s close colleagues within and outside the same team in the relationship between rapport-building behavior and empathy for colleagues.
In previous studies, the focus on rapport-building behavior was limited to the relationship between the employee and the customer. However, this study makes a theoretical contribution as well, in that it broadens the target group for research by focusing on the rapport-building behavior among cabin crew members, which is an extension of previous research that focused only on the relationship between the employee and the customer.
No previous research exists on rapport-building behavior and empathy among cabin crew members. Therefore, this study is unique in that it is the first to set cabin crew members as the target group for research. In addition, it is academically significant as it analyzed which of the five factors of rapport-building behavior had the strongest effect on engendering empathy among colleagues. It differs from previous studies in that it concludes that if the positive interaction between crew members continues, then effects of individual, psychological, economic benefits, and social benefits may be achieved from the perspective of cabin crew members as well as the company. This finding will be the cornerstone for human resource management planning and its improvement; in the creation of a positive work environment; and in the improvement of mental health and quality of life, work–life balance, and organizational management.
2.Second, In introduction, please provide implications of the current research -both theoretically and practically - and originality clearly.
Response
Thank you for your insightful comments and helpful suggestions. In accordance with your suggestions, we have clearly presented the implications and originality of the study in the Introduction. In addition, we have revised the Introduction for re-submission, as presented below.
- Introduction
To date, in South Korea, there has been a constant interest regarding ways to improve airline cabin crews’ performance and the services offered to the customers. However, most recent studies have focused on ensuring the human rights of individual cabin crew members and on other psychological and emotional aspects of the job [1-3].
The purpose of this study is to confirm the effectiveness of rapport-building behavior among cabin crew members in enhancing their cooperation with each other and increasing the synergy effect achieved.
A literature review revealed that rapport-building behavior strengthens the psychological trust between the service provider and the customer, and additionally, improves the quality of interaction between them [4]. Furthermore, the communication between the leader and members, i.e., "leader–member" communication, is viewed as an extremely important factor. Therefore, a method that leaders can utilize to improve the quality of the relationship was presented, and a model of leadership communication was suggested based on the rapport management theory [5]. Additionally, there are studies that report that the more empathetic the leader is, the higher the member’s job satisfaction is likely to be [6]. Therefore, this study expanded on previous research and focused on the relationship among cabin crew members. However, few studies have focused on rapport-building behaviors and empathy among cabin crew colleagues. In this study, we considered aspects such as airline workers’ working conditions and dignity of labor and expanded the concepts of rapport and empathy from the context of basic human relationships to the context of airline cabin crew members.
Cabin crew members in Korea's domestic airlines work as a team. A few previous studies show that psychological bonds are important for rapport building, which is crucial for cabin crew members who always work in close cooperation. Therefore, this study also focused on rapport-building behaviors.
Based on a comprehensive literature review, we classified rapport-building behavior between cabin crew members into five categories: (1) uncommonly attentive behavior between the crew members, (2) common-grounding behavior, (3) courteous behavior, (4) connecting behavior, and (5) information-sharing behavior [7-10]. We then investigated the behaviors that had the strongest effect on empathy among the crew members.
Because a cabin crew works within a group/team, a psychological bond between colleagues is essential to help the members intuitively identify instances when their colleagues experience difficulties during their work and the type of assistance they may require. Academics have defined this type of psychological bond as empathy [11-13]. Empathy refers to a strong, deep psychological connection between two individuals, which can be formed through smooth communication (rapport building) [14,15]. In this study, it was found that rapport-building behavior has a positive effect on empathy.
Previous studies have reported that strong empathy with colleagues can have a positive effect on team performance [16-18], organizational atmosphere [19-21] while preventing irregularity among employees [22-24]. In this study, we investigated the causality of the relationship between empathy and team performance, organizational atmosphere, and employee irregularity. We also performed a statistical analysis of the mediating role of having one’s close colleagues within and outside the same team in the relationship between rapport-building behavior and empathy for colleagues.
In previous studies, the focus on rapport-building behavior was limited to the relationship between the employee and the customer. However, this study makes a theoretical contribution as well, in that it broadens the target group for research by focusing on the rapport-building behavior among cabin crew members, which is an extension of previous research that focused only on the relationship between the employee and the customer.
No previous research exists on rapport-building behavior and empathy among cabin crew members. Therefore, this study is unique in that it is the first to set cabin crew members as the target group for research. In addition, it is academically significant as it analyzed which of the five factors of rapport-building behavior had the strongest effect on engendering empathy among colleagues. It differs from previous studies in that it concludes that if the positive interaction between crew members continues, then effects of individual, psychological, economic benefits, and social benefits may be achieved from the perspective of cabin crew members as well as the company. This finding will be the cornerstone for human resource management planning and its improvement; in the creation of a positive work environment; and in the improvement of mental health and quality of life, work–life balance, and organizational management.
3.Third, Theoretical evidence to support hypotheses are lack. Focus on the underlying mechanism for each hypothesis, and please provide them.
Response
In accordance with your suggestions, we have incorporated additional theoretical evidence to support the hypotheses to compensate for the parts previously considered lacking.
1.2. Literature Review
1.2.1. Rapport-building behavior
The word rapport originated from French and refers to a relationship of trust between people, It is a comprehensive term for synchronization, wherein mutual communication transcends language, and occurs at the physiological or psychological level and also entails a mirror effect, where individuals subconsciously mimic one another’s behavior [20,21].
After Freud, several authors have offered their own definitions of rapport, particularly in the business context. For example, rapport could imply a perception of mutual connection between staff and clients [22], or an intimate relationship formed on the basis of trust while providing a service [23]. One author defined rapport as the relationship between a service provider and a client based on experiencing positive, amicable emotions from mutual interactions [24], while others have described it as a relationship that creates an atmosphere of mutual understanding and trust [10], and a behavioral element aimed at improving the quality of the relationships between service providers and clients [25]. Drawing on the common elements from previous definitions, this study defines rapport as the formation of trust, camaraderie, and a psychological relationship between the members of an airline cabin crew interacting amongst each other.
Rapport is an important factor affecting economic outcomes, as developing rapport can improve customer loyalty [26–29]. At an inter-personal level, rapport-building involves increasing comfort and intimacy by expressing attention through various ways, such as eye contact, control of one’s breathing, posture, movements, and other forms of body language. However, social rapport has a positive effect on the satisfaction of other parties [30].
Previous studies have classified rapport-building behavior into five types: uncommonly attentive behavior, common grounding behavior, courteous behavior, connecting behavior, and information-sharing behavior [2–5]. In this study, we analyzed these behavioral types to measure the rapport-building behavior of cabin crew members with their colleagues.
Previous studies have also shown that rapport-building behavior can create empathy among colleagues. For instance, information-sharing behavior and common grounding behavior by tour guides have been shown to have a positive effect on clients’ empathy [4], while the use of rapport-building techniques during interpersonal conflict has positive effects on empathy and the willingness to resolve the conflicts [10]. In addition, several studies have reported that personalization and listening are among the four effective rapport-building techniques, which positively impact rapport building between female investigators and government officials [14]. Accordingly, the following subsections outline the behavioral types and presents our hypotheses.
Reference
Kim, S.U. Rapport building in investigative interviewing by using four rapport building techniques. Korean J. Cult. Soc. Issues. 2012, 19, 487–506
1.2.2 Empathy toward colleagues
Empathy can be divided into (a) cognitive empathy, which is the process of understanding others’ behavior and adopting their roles, and (b) emotional empathy, which entails experiencing others’ emotions vicariously [46]. Empathy enhances one’s understanding of others and is an essential factor for forming positive interpersonal relationships [47]; therefore, for an airline cabin crew, it is a core element of relationships among colleagues.
Empathy can be evoked in various work interactions, such as when a colleague expresses emotions about a difficult problem, while looking after a colleague, acting gently, etc. [48–50]. Colleagues responding to each other’s problems exemplifies empathy within the group [51]. Being an essential part of relationship-building, empathy also increases intimacy among colleagues [52]. Moreover, empathy toward colleagues promotes a positive transformation of negative emotional experiences that occur within a group [53,54]. Empathy also increases group involvement and positive emotions and attitudes while reducing personal distress [55].
Many studies have reported that empathy toward one’s colleagues significantly impacts organizational effectiveness, whereas empathy from leaders positively affects the performance and failure tolerance of their subordinates. It also promotes group cohesion and work satisfaction [56]. Additionally, several studies have reported a positive correlation between the leader’s empathic ability and the member’s job satisfaction [6].
However, few studies have focused on airline cabin crew members’ perception and experience of empathy from their colleagues. Therefore, to fill this gap, this study provides a thorough analysis of cabin crew members’ experiences of empathy within and outside their in-groups to understand its impact on various performance measures and the overall group environment.
Reference
Shim,H.J. Correlation between Superiors’ Empathy Ability Perceived and Job Satisfaction According to Office Workers’ Sex and Adult Attachment. Master’s Thesis, Hongik University Graduate School, Seoul, Korea,2014.
1.2.3 Team performance
Team performance pertains to the results obtained using multiple resources in pursuit of a team goal [57].
Like in other countries, full-service carriers in South Korea manage their cabin crew in team/group units. Team performance is scored by combining the individual performances of cabin crew members, and the scores are compared to other teams to obtain a team unit score. These results are then reflected in the performance evaluation of individual team members [58].
Teamwork is extremely important for airline cabin crews. For example, a team with good bond between individual team members is better at achieving its goals, which is also reflected in the team’s performance levels [59].
Numerous studies have reported the positive effect of empathy toward colleagues on team performance. If the leader is empathetic, members can easily overcome failures and become motivated, which will have a positive effect on the overall performance of the organization [16]. Several studies have reported that empathy among colleagues has a positive impact on the team/group performance [12]. Moreover, airline cabin crews with high empathic ability show stronger attachment to their working group, stronger organizational involvement, and better cooperation between colleagues; empathic ability also reduces workplace stress and is closely related to psychological comfort [13]. Therefore, we proposed the following hypothesis:
Hypothesis 6 (H6): Cabin crew with high empathy among colleagues show better team performance.
Reference
Moh, Y.H.; Lee, C.W.; Kim, K.S. The effects of a leader’s compassion on organization members’ performance: an investigation of the mediating effect of organization members’ failure tolerance. Korean Manag. Consulting Rev. 2020, 20, 25–37.
1.2.5 Irregularity
In organizations, accidents occur due to a combination of factors, and the number of accidents can be reduced by controlling or eliminating its contributing factors [17]. In the present study, we investigated irregularities resulting from human factors that lie outside the environmental and physical factors. In the airline industry, human factors that cause irregularities pertain to mistakes by the cabin crew, cockpit crew, maintenance staff, ground crew, and air traffic controllers [67].
Human error by airline staff contributes to over 70% of aircraft accidents worldwide [68]. To prevent these accidents, several airlines have developed crew resource management (CRM) training programs as part of their safety education training for airline workers, which are used in airlines across the globe [69].
Irregularities from human errors by the cabin crew can manifest as mistakes or carelessness by the cabin crew; inaccurate situational awareness; tension within the work atmosphere; poor working condition due to work stress; insufficient partnership and emotional friction among colleagues; insufficient communication and lack of information sharing and feedback between the cabin crew; as well as delays in service due to teamwork failure [70].
Studies on the effects of empathy and irregularity have reported that the perceived fatigue within an airline cabin crew affects irregularity [18] and that communication levels within an organization, empathy, and a safe atmosphere reduced the rate of industrial accidents by safety and public health administrators [71]. In addition, during cabin crew briefing, the better the information that is communicated or the more focused the voice, the more effective it is in preventing accidents [22].
Based on these studies, we consider empathy toward colleagues as a factor that prevents irregularities within an organization. Therefore, we proposed the following hypothesis:
Hypothesis 8 (H8): Cabin crews with members who show greater empathy towards colleagues are more effective in preventing irregularities and can help airlines take retroactive measures for long-term safety.
Reference
Oh, J.A.; Hyun, S.H.; Jeong, J.Y. A study on the causal relation analysis between the irregularity and work performance in the influence of briefing style. Korean. J. Hosp. Tourism. 2019, 28, 293–308.
4.Fourth, H9 - H13 (moderating hypotheses) would be better if it is more specific (such as, focusing on how different?)
Response
In accordance with your suggestions, the part explaining Hypotheses 9–13 on the moderator variable was supplemented with further explanation.
1.2.6 Moderating effects of a cabin crew’s rapport-building behavior within and outside their in-group
Airlines manage and supervise cabin crews using a team-based system that places high emphasis on teamwork and efficient human management.
Since 1997, Asiana Airlines follows a group management system, where each group consists of one group leader and 11–13 cabin crew members. Asiana Airlines comprises of three teams [72].
Korean Air has implemented a team system since 1986, with 390 groups of cabin crew, where each team on an average is composed of a team leader, team vice-leader, and 13–15 team members, and the teams are maintained for one year at a time [73].
Team members inevitably improve their feelings of togetherness, intimacy, and bonding as they spend more time with other members while working on flights or away from work. However, there has been a shortage of research on rapport-building behaviors or empathy among the members of a cabin crew.
A multi-group analysis was performed to test whether there were any significant differences in the effects of rapport-building behaviors on empathy toward colleagues, moderated by whether the participants’ closest colleagues worked in the same team.
The path coefficients and their significance were investigated even though the paths from rapport-building behaviors to empathy toward colleagues showed no difference for the moderating role of the presence of the participants’ closest colleagues within the same team.
The results show that the effects of the four rapport-building behaviors were relatively evenly distributed when the participants’ closest colleagues were in the same team, whereas uncommonly attentive behavior had a stronger effect than other types of rapport-building behaviors in the case that the participants’ closest colleagues were not in the same team.
Therefore, we propose the following hypotheses:

Reviewer 2 Report
Every chapter 1 needs to be better presented. It is not structured as a single chapter. The array appears at the end without any presentation. It is possible to understand, but it is necessary to improve the presentation of the structure of the chapter, in order to value all the work of the authors.
Author Response
Thank you for your insightful comments and helpful suggestions. We have thoroughly reviewed your comments and addressed them to the best of our abilities to improve the quality of the paper.
1.Every chapter 1 needs to be better presented. It is not structured as a single chapter. The array appears at the end without any presentation. It is possible to understand, but it is necessary to improve the presentation of the structure of the chapter, in order to value all the work of the authors.
Response
In accordance with your suggestions, Chapter 1 has been logically re-arranged and the Introduction rewritten. The purpose and implications of the study, the current status, and the background of the study were revised.
- Introduction
To date, in South Korea, there has been a constant interest regarding ways to improve airline cabin crews’ performance and the services offered to the customers. However, most recent studies have focused on ensuring the human rights of individual cabin crew members and on other psychological and emotional aspects of the job [1-3].
The purpose of this study is to confirm the effectiveness of rapport-building behavior among cabin crew members in enhancing their cooperation with each other and increasing the synergy effect achieved.
A literature review revealed that rapport-building behavior strengthens the psychological trust between the service provider and the customer, and additionally, improves the quality of interaction between them [4]. Furthermore, the communication between the leader and members, i.e., "leader–member" communication, is viewed as an extremely important factor. Therefore, a method that leaders can utilize to improve the quality of the relationship was presented, and a model of leadership communication was suggested based on the rapport management theory [5]. Additionally, there are studies that report that the more empathetic the leader is, the higher the member’s job satisfaction is likely to be [6]. Therefore, this study expanded on previous research and focused on the relationship among cabin crew members. However, few studies have focused on rapport-building behaviors and empathy among cabin crew colleagues. In this study, we considered aspects such as airline workers’ working conditions and dignity of labor and expanded the concepts of rapport and empathy from the context of basic human relationships to the context of airline cabin crew members.
Cabin crew members in Korea's domestic airlines work as a team. A few previous studies show that psychological bonds are important for rapport building, which is crucial for cabin crew members who always work in close cooperation. Therefore, this study also focused on rapport-building behaviors.
Based on a comprehensive literature review, we classified rapport-building behavior between cabin crew members into five categories: (1) uncommonly attentive behavior between the crew members, (2) common-grounding behavior, (3) courteous behavior, (4) connecting behavior, and (5) information-sharing behavior [7-10]. We then investigated the behaviors that had the strongest effect on empathy among the crew members.
Because a cabin crew works within a group/team, a psychological bond between colleagues is essential to help the members intuitively identify instances when their colleagues experience difficulties during their work and the type of assistance they may require. Academics have defined this type of psychological bond as empathy [11-13]. Empathy refers to a strong, deep psychological connection between two individuals, which can be formed through smooth communication (rapport building) [14,15]. In this study, it was found that rapport-building behavior has a positive effect on empathy.
Previous studies have reported that strong empathy with colleagues can have a positive effect on team performance [16-18], organizational atmosphere [19-21] while preventing irregularity among employees [22-24]. In this study, we investigated the causality of the relationship between empathy and team performance, organizational atmosphere, and employee irregularity. We also performed a statistical analysis of the mediating role of having one’s close colleagues within and outside the same team in the relationship between rapport-building behavior and empathy for colleagues.
In previous studies, the focus on rapport-building behavior was limited to the relationship between the employee and the customer. However, this study makes a theoretical contribution as well, in that it broadens the target group for research by focusing on the rapport-building behavior among cabin crew members, which is an extension of previous research that focused only on the relationship between the employee and the customer.
No previous research exists on rapport-building behavior and empathy among cabin crew members. Therefore, this study is unique in that it is the first to set cabin crew members as the target group for research. In addition, it is academically significant as it analyzed which of the five factors of rapport-building behavior had the strongest effect on engendering empathy among colleagues. It differs from previous studies in that it concludes that if the positive interaction between crew members continues, then effects of individual, psychological, economic benefits, and social benefits may be achieved from the perspective of cabin crew members as well as the company. This finding will be the cornerstone for human resource management planning and its improvement; in the creation of a positive work environment; and in the improvement of mental health and quality of life, work–life balance, and organizational management.
Reference:
1.Yu, M.J.; Hyun, S.H. Development of Modern Racism Scale in Global Airlines: A Study of Asian Female Flight Attendants. Int J.Environ. Res.Public Health.2021,18,2688
2.Song, M.N.; Choi, H.J.; Hyun, S.H. MBTI Personality Types of Korean Cabin Crew in Middle Eastern Airlines and Their Associations with Cross-Cultural Adjustment Competency, Occupational Competency, Coping Competency, Mental Health and Turnover Intention. Int. J. Environ. Res. Public Health.2021, 18, 3419-3438.
3.Kim, H.L.; Hyun, S.H. Developing a Stigma Scale for the Workplace: Focus on an Airline Cabin Crew. Int.J. Environ. Res. Public Health. 2021,18, 4003-4022.
4.Ji, S.G.; Yang, B.S.; Kim, S.H. The effects of rapport of healthcare services providers on emotional labor, job satisfaction and organizational commitment. J. Korea Serv. Manag. Soc. 2010, 11, 209–236.
5.Campbell, K. S.; White, C. D.; Johnson, D. E. Leader-member relations as a function of rapport management.J. Business Commun.2003,40,170-194.
- Shim,H.J. Correlation between Superiors’ Empathy Ability Perceived and Job Satisfaction According to Office Workers’ Sex and Adult Attachment. Master’s Thesis, Hongik University Graduate School, Seoul, Korea,2014.

Reviewer 3 Report
- The paper could benefit from professional editing for grammar and expression.
- The notion of rapport is, more often than not, applied to relationship between service providers and clients. This paper refers to rapport among team members, which is not the popular notion. It is recommended that more previous works are cited to fill this gap, such as: Campbell, K. S., White, C. D., & Johnson, D. E. (2003). Leader-member relations as a function of rapport management. The Journal of Business Communication (1973), 40(3), 170-194.
- In line with the above comment, some sections in the literature review (actually, all of the behaviors except for common grounding) are focused on service provider-client relationship. Rewriting these so that the focus is on the team member relationship is required.
- The small sample size should be listed as a limitation
- Were there any differences in ranks and responsibilities of the respondents?
- Another major limitation is that this study is quite distant from the more popular stream of research on team member exchange. Please contemplate on how you can better link this study to the mainstream works.
Author Response
Thank you for your insightful comments and helpful suggestions. We have thoroughly reviewed your comments and addressed them to the best of our abilities to improve the quality of the paper.
1.The paper could benefit from professional editing for grammar and expression.
Response
In accordance with your suggestions, we have reviewed the grammar and the overall expression of the paper and have meticulously revised the document.
2.The notion of rapport is, more often than not, applied to relationship between service providers and clients. This paper refers to rapport among team members, which is not the popular notion. It is recommended that more previous works are cited to fill this gap, such as: Campbell, K. S., White, C. D., & Johnson, D. E. (2003). Leader-member relations as a function of rapport management. The Journal of Business Communication (1973), 40(3), 170-194.
Response
Thank you very much for providing us the references. they were extremely useful. In accordance with your suggestions, we have added these references and have re-written the Introduction. We found your opinion to be highly valuable, as a result of which the contents of the research were also added throughout our paper. Thank you once again.
- Introduction
To date, in South Korea, there has been a constant interest regarding ways to improve airline cabin crews’ performance and the services offered to the customers. However, most recent studies have focused on ensuring the human rights of individual cabin crew members and on other psychological and emotional aspects of the job [1-3].
The purpose of this study is to confirm the effectiveness of rapport-building behavior among cabin crew members in enhancing their cooperation with each other and increasing the synergy effect achieved.
A literature review revealed that rapport-building behavior strengthens the psychological trust between the service provider and the customer, and additionally, improves the quality of interaction between them [4]. Furthermore, the communication between the leader and members, i.e., "leader–member" communication, is viewed as an extremely important factor. Therefore, a method that leaders can utilize to improve the quality of the relationship was presented, and a model of leadership communication was suggested based on the rapport management theory [5]. Additionally, there are studies that report that the more empathetic the leader is, the higher the member’s job satisfaction is likely to be [6]. Therefore, this study expanded on previous research and focused on the relationship among cabin crew members.
Reference:
[4] Ji, S.G.; Yang, B.S.; Kim, S.H. The effects of rapport of healthcare services providers on emotional labor, job satisfaction and organizational commitment. J. Korea Serv. Manag. Soc. 2010, 11, 209–236.
[5] Campbell, K. S.;White, C. D.; Johnson, D. E. Leader-member relations as a function of rapport management.J. Business Commun.2003,40,170-194.
[6] Shim,H.J. Correlation between Superiors’ Empathy Ability Perceived and Job Satisfaction According to Office Workers’ Sex and Adult Attachment. Master’s Thesis, Hongik University Graduate School, Seoul, Korea,2014.
3.In line with the above comment, some sections in the literature review (actually, all of the behaviors except for common grounding) are focused on service provider-client relationship. Rewriting these so that the focus is on the team member relationship is required.
Response
Most of the previous research focuses on the relationship between the service provider and the consumer (or customer). This limited relationship was extended and applied to cabin crew members working in the field, as a result of which some sections of the literature review were supplemented and re-written.
1.2.1.1 Uncommonly attentive behavior
Uncommonly attentive behaviors can be divided into three subtypes [31]. The first subtype is an atypical behavior, which involves making the other party recognize that they are receiving an unexpected special treatment [2]. The second subtype is personalized rapport-building, which stems from remembering the name or personal information of a specific client [32]. The third subtype is sincere personal attention [3]. For example, cabin crew members may make it a point to remember their colleagues' birthdays and serve birthday cakes at overseas stations. When they meet on the next flight, they may remember their colleagues’ names and present them with drinks or gifts. Therefore, we hypothesized:
Hypothesis 1 (H1): Uncommonly attentive behavior by cabin crew members would make them more empathetic toward their colleagues.
1.2.1.2 Common grounding behavior
Common grounding behaviors occur when individuals try to find mutual areas of interest. They can be divided into two subtypes [33]. The first involves looking for similarities with a coworker regardless of differences in sales or performance levels. Two parties who identify their similarities and perceive each other as being similar tend to build rapport and make their relationship more robust [34,35]. The second involves the act of looking for mutual interest, beyond superficial similarities [36]. For example, there may be such colleagues who receive the initial training together, advanced customer service training together, or return-to-work training together following maternity leave. In addition, there are several in-house clubs and societies where acquaintances may be formed based on common interests, which may blossom into lifelong reliable relationships. Hence, we proposed:
Hypothesis 2 (H2): Common grounding behavior by cabin crew members would make them more empathetic toward their colleagues.
1.2.1.3 Courteous behavior
Courteous behavior implies earnestly treating the other party with interest and respect. Choi’s study showed that exhibiting courteous behavior aided rapport building [37].
Courteous behavior can be divided into three subtypes: the act of sharing empathy with a client, which is considered to be an important factor in increasing intimacy and rapport building [32,38]; having respectful manners [39–41]; and honesty, which includes displaying a friendly attitude and greeting the client. During problematic scenarios, sincere apologies and behaviors aimed at restoring the relationship have been claimed to induce positive rapport building [42]. For example, cabin crew members are courteous toward their colleagues. They may greet each other with bright smiles in pre-flight briefing rooms, nod as they walk past each other in the plane aisle, and exchange warm greetings in the galley during flights. Hence, we proposed:
Hypothesis 3 (H3): Courteous behavior by cabin crew members would make them more empathetic toward their colleagues.
1.2.1.4 Connecting behavior
Connecting behavior relies on setting clear intention to develop a relationship [43], which is exemplified through actions such as pleasant and humorous conversations and speaking on familiar terms [44]. Connecting behavior can be divided into three subtypes [2]. First is a pleasant conversation with a client. The second is speaking on familiar terms. Connecting behavior that creates an atmosphere of entertainment or interest has been reported to help form strong bonds [2]; therefore, the third subtype involves humor [31]. For example, as cabin crew members work in teams or groups, they may hold humorous conversations with new colleagues. This will help lighten the mood and facilitate quick friendships, thereby building rapport in short time. This will, in turn, be helpful in their in-flight work requiring cooperation. Hence, we proposed:
Hypothesis 4 (H4): Connecting behavior by cabin crew members would make them more empathetic toward their colleagues.
1.2.1.5 Information-sharing behavior
Information-sharing behavior includes behaviors such as providing clear information, sharing knowledge, and providing overall advice about a service [2]. Information-sharing behavior makes the client feels that they are receiving a differentiated service, and both the service provider and client form a relationship based on amicable emotional exchange [45]. As part of this behavior, service provides establishes open communication with the client [43], which also improves transparency. For example, cabin crew members may share their colleagues’ latest flight information and predict service items on their flight routes in advance. They may also share information on pre-flight preparation, exercise management, and even overseas travel knowledge. They may share information on the company's latest announcements, and may even study together for regular safety training sessions. Hence, we proposed:
Hypothesis 5 (H5): Information-sharing behavior by cabin crew members would make them more empathetic toward their colleagues.
4.The small sample size should be listed as a limitation
Response
In accordance with your suggestions, we have added the remarks about the small sample size in the section about limitations and supplemented the section.
- Limitation and Recommendation for Future Research
This study has several limitations, based on which we suggest some direction for future research to address these limitations.
First, during the study’s survey period, the airline industry was in the midst of a serious management crisis due to the COVID-19 pandemic, which at the time of the research was threatening very existence of the airline industry. During the survey, many cabin crew members were on an unpaid or rotational leave. This disconnect from the direct experience of the working environment could have invariably affected the results of the survey. A comparative study by repeating the survey after the airline industry had recovered and the cabin crew had returned to normal work schedules would provide better insights on the accuracy of the results.
Second, the moderating effect of the presence of the participants’ closest colleagues within the same team on the relationship between rapport-building behaviors and empathy among cabin crew colleagues was rejected. The COVID-19 pandemic has led to a sustained decrease in demand in the airline industry, and the cutbacks in the number of flights and personnel have led to an increase in the number of cabin crews on leave. Currently, among the domestic full-service carriers, Asiana Airlines has been acquired by Korean Air, which are soon planned to merge. Due to staffing reductions, there has been a serious lack of flight work for the airline crew. Thus, the shortage of flights to investigate the differences between cabin crew members on the same/different teams meant that the moderating variable was not fully represented by the measured survey items.
Third, the sample mostly consisted of cabin crew working for domestic full-service carriers (Korean Air, Asiana Airlines). Therefore, future studies could expand the framework of this study to include cabin crew members from diverse low-cost carriers and overseas airlines to improve the generalizability of the results.
Fourth, a broader and larger sample could not be obtained because of social distancing and other environmental constraints, given that the survey was conducted after the onset of the COVID-19 pandemic. Thus, it is difficult to generalize the research results owing to the small sample size. It is imperative to collect more samples for greater result reliability of future studies.
Fifth, the notion of rapport has been studied in the consumers behavior field. This study expanded the existing study into human resource area. For the future study, it is necessary to link the rapport with the mainstream of human resource areas, such as leader-member exchange, employee work stress, turnover rate, and employee well-being.
5.Were there any differences in ranks and responsibilities of the respondents?
Response
In accordance with your suggestions, we checked for potential differences in ranks and responsibilities of the respondents. However, we found no such differences. We thank you for your suggestions and remain open to other ideas or suggestions on data analysis.
6.Another major limitation is that this study is quite distant from the more popular stream of research on team member exchange. Please contemplate on how you can better link this study to the mainstream works.
Response
Thank you for your valuable suggestion, in accordance with which we have inserted a new paragraph, as shown below, in the section titled: 6. Limitation and Recommendation for Future Research.
- Limitation and Recommendation for Future Research
This study has several limitations, based on which we suggest some direction for future research to address these limitations.
First, during the study’s survey period, the airline industry was in the midst of a serious management crisis due to the COVID-19 pandemic, which at the time of the research was threatening very existence of the airline industry. During the survey, many cabin crew members were on an unpaid or rotational leave. This disconnect from the direct experience of the working environment could have invariably affected the results of the survey. A comparative study by repeating the survey after the airline industry had recovered and the cabin crew had returned to normal work schedules would provide better insights on the accuracy of the results.
Second, the moderating effect of the presence of the participants’ closest colleagues within the same team on the relationship between rapport-building behaviors and empathy among cabin crew colleagues was rejected. The COVID-19 pandemic has led to a sustained decrease in demand in the airline industry, and the cutbacks in the number of flights and personnel have led to an increase in the number of cabin crews on leave. Currently, among the domestic full-service carriers, Asiana Airlines has been acquired by Korean Air, which are soon planned to merge. Due to staffing reductions, there has been a serious lack of flight work for the airline crew. Thus, the shortage of flights to investigate the differences between cabin crew members on the same/different teams meant that the moderating variable was not fully represented by the measured survey items.
Third, the sample mostly consisted of cabin crew working for domestic full-service carriers (Korean Air, Asiana Airlines). Therefore, future studies could expand the framework of this study to include cabin crew members from diverse low-cost carriers and overseas airlines to improve the generalizability of the results.
Fourth, a broader and larger sample could not be obtained because of social distancing and other environmental constraints, given that the survey was conducted after the onset of the COVID-19 pandemic. Thus, it is difficult to generalize the research results owing to the small sample size. It is imperative to collect more samples for greater result reliability of future studies.
Fifth, the notion of rapport has been studied in the consumers behavior field. This study expanded the existing study into human resource area. For the future study, it is necessary to link the rapport with the mainstream of human resource areas, such as leader-member exchange, employee work stress, turnover rate, and employee well-being.

Round 2
Reviewer 1 Report
Thank you for your work on the revision.